# Peer Support and Mental Health of Migrant Domestic Workers: A Scoping Review

**DOI:** 10.3390/ijerph19137617

**Published:** 2022-06-22

**Authors:** Ken Hok Man Ho, Chen Yang, Alex Kwun Yat Leung, Daniel Bressington, Wai Tong Chien, Qijin Cheng, Daphne Sze Ki Cheung

**Affiliations:** 1The Nethersole School of Nursing, Faculty of Medicine, The Chinese University of Hong Kong SAR, Shatin, N.T., Hong Kong SAR, China; chenyang@cuhk.edu.hk (C.Y.); wtchien@cuhk.edu.hk (W.T.C.); 2Department of Social Work, The Chinese University of Hong Kong, Shatin, N.T., Hong Kong SAR, China; alexkyleung@cuhk.edu.hk (A.K.Y.L.); qcheng@cuhk.edu.hk (Q.C.); 3College of Nursing and Midwifery, Charles Darwin University, Darwin 0815, Australia; daniel.bressington@cdu.edu.au; 4School of Nursing, The Hong Kong Polytechnic University, Hung Hom, Kowloon, Hong Kong SAR, China; daphne.cheung@polyu.edu.hk

**Keywords:** mental health, migrant domestic worker, peer support, scoping review

## Abstract

The effectiveness of peer support in improving mental health and well-being has been well documented for vulnerable populations. However, how peer support is delivered to migrant domestic workers (MDWs) to support their mental health is still unknown. This scoping review aimed to synthesize evidence on existing peer support services for improving mental health among MDWs. We systematically searched eight electronic databases, as well as grey literature. Two reviewers independently performed title/abstract and full-text screening, and data extraction. Twelve articles were finally included. Two types of peer support were identified from the included studies, i.e., mutual aid and para-professional trained peer support. MDWs mainly seek support from peers through mutual aid for emotional comfort. The study’s findings suggest that the para-professional peer support training program was highly feasible and culturally appropriate for MDWs. However, several barriers were identified to affect the successful implementation of peer support, such as concerns about emotion contagion among peers, worries about disclosure of personal information, and lack of support from health professionals. Culture-specific peer support programs should be developed in the future to overcome these barriers to promote more effective mental health practices.

## 1. Introduction

Migrant domestic workers (MDWs) are one of the largest migrant populations around the world. A MDW is a full-time live-in migrant worker who is tied to an employer either through a regulated work permit or underground contract that allows the MDWs to work for a single household [1]. Globally, there are no universal job specifications or proficiency requirements for MDWs. They typically perform both caring work and domestic work, known as the three Cs (i.e., cooking, cleaning, and caring) for children or older adults [2]. In the era of global population aging, emerging evidence has showed that living with a MDW is associated with the better mental well-being of older adults (e.g., lower level of loneliness) [3,4]. In 2015, there were around 11 million MDWs globally [5], which were mainly distributed in Arab States (27.4%); Eastern Asia (9.5%); Northern, Southern, and Western Europe (19.2%); South-Eastern Asia and the Pacific (19.4%) [6]. Over 70% of MDWs were women [5]. The duration of stay in host countries varies according to the employment contract (e.g., 2 years in Hong Kong) or rights to permanent residency (e.g., Canada) and there are no international or local statistics about the average length of stay of MDWs. Internationally, MDWs represent a great vulnerable and marginalized working group, which is characterized by excessive demands, low skill, and limited autonomy [7,8,9]. The majority of MDWs come from low- and middle-income countries, and are exposed to discrimination, exploitation, and abuse in host countries [10]. According to the International Labor Organization [11], MDWs are usually not protected under labor laws of host countries. They had inadequate health insurance for medical diseases in host countries compared to local populations. Due to cultural and language barriers, they are easily marginalized from the health and social care system of the host countries [10]. All of these factors leave them vulnerable to mental health problems.

A number of quantitative studies suggest that common mental health problems in MDWs include depression, anxiety, loneliness, and stress [12,13,14,15,16]. Estimates from a recent review show that 10.3–18.2% of MDWs suffered from moderate to severe levels of depression across countries [10], which was comparable to that of the general population in Hong Kong (19% had depression as measured by the patient health questionnaire-9) during the COVID-19 pandemic [17]. A cross-sectional study involving 346 MDWs in Hong Kong found that nearly 40% of them were at risk for high psychological distress, as measured by the 14-item Hospital Anxiety and Depression Scale [15]. Another cross-sectional study in Singapore found that over half of female MDWs reported being stressed, contributing to a worse quality of life [18]. Meanwhile, a survey in Taiwan showed that 16.5% of Indonesian MDWs had depressive symptoms [14]. Another study in Macao including 1375 Filipino MDWs found that 38% and 32% of them met the criteria for depression and anxiety, respectively [19]. Throughout the COVID-19 pandemic, MDWs usually lacked adequate COVID-19-related information [20] and sufficient protective equipment [21], and had much less access to local healthcare services [22], which negatively impacted their mental health. Particularly, their live-in status exacerbated their vulnerabilities because some employers forbade MDWs from leaving home during the COVID-19 pandemic [23]. The deterioration of mental health and well-being can result in loss of social functioning, hospitalizations, and even suicide [18]. However, the healthcare system of host countries has long overlooked the issues in mental health among MDWs; thus, establishing services supporting their mental health and well-being is important.

Maintaining and strengthening social networks are believed to be effective approaches to protect MDWs and other vulnerable groups against mental health problems [10]. Connections with family members and friends have been the commonest means for MDWs to seek emotional support and relieve stress [10,24]. Due to physical and emotional separation from their loved ones living in countries of origin, support from MDWs’ peers who share common beliefs, ethnicity, and culture occupies a significant part of their social network in the host country. Indeed, it is commonly acknowledged that peer support within a collectivist society is critical for stress coping and mental health support [25]. As such, peer support, which has been widely used in social work, psychological counselling, and healthcare services, may be well suited to improve the mental health of the MDWs [26].

Within the healthcare context, peer support is defined as “the provision of emotional, appraisal, and informational assistance by a created social network member who possesses experiential knowledge of a specific behavior or stressor and similar characteristics as the target population” [27]. People with similar experiences and stressors are brought together to share individual experiences and provide reciprocal help on the basis of respect, trust, and mutual agreement [26,28]. It provides a platform for individuals to be connected with other peers and their community and allows an individual to engender a collective sense of belonging and connectedness to a group with common concerns and needs [26,29]. Through this process, recipients of peer support can gain knowledge of coping strategies for their psychological problems, as well as build confidence and personal resources for improving mental health. In addition, peer support can also enhance psychological resilience, which allows an individual to be able to “adapt to the challenges of life and maintain mental health despite exposure to adversity” [30]. Higher psychological resilience exerts a protective effect to stressful events [31] so as to protect individuals from mental health problems.

Peer support has received increasing attention in mental health research. A meta-analysis showed that peer support intervention is more effective than usual care in reducing depression symptoms in postpartum women, mothers of school-aged children, HIV-positive men, and patients with stage II cancer, with a medium pooled effect size (SMD = −0.59) [32]. Several peer support programs have been developed to promote mental health and well-being in disadvantaged groups, such as migrants [33,34] and refugees [35,36,37], in great need of support. Peer support has also been advocated by the World Health Organization (WHO) as an important approach to increase well-being and mitigate negative effects of social isolation and stressors associated with migration [38]. It has been regarded as an alternative or a complement to the formal mental healthcare system/service [39].

Although the benefits of peer support in mental healthcare have been established in postpartum women, mothers of school-aged children, HIV-positive men, patients with stage II cancer, and refugees, little is known about its applications for supporting MDWs. Their cultural and linguistic diversity, marginalization from health and social care, and specific work-related stressors (e.g., live-in arrangement) may pose great challenges for the design and implementation of peer support intervention in this population. Given the growing number of MDWs worldwide and the popularity of mental health problems among this vulnerable group, there is a prominent need to better understand the importance and relevant factors influencing peer support before its use in MDWs for their mental wellbeing. Therefore, we performed a scoping review to map currently available information or evidence of peer support for promoting the mental health of MDWs, and to provide recommendations on applying peer support into mental health promotion and intervention.

## 2. Methods

### 2.1. Study Design

This review followed the methodological framework for scoping reviews outlined by Arksey and O’Malley [40] and expanded by Levac et al. [41], involving (1) identifying the research questions; (2) identifying relevant information; (3) study selection; (4) charting the data; (5) collating, summarizing, and reporting the results. The protocol of this review has been registered at the Open Science Framework (Registration DOI: 10.17605/OSF.IO/SNVAW).

### 2.2. Identifying the Research Questions

The main research questions were: (1) “what types of peer support are available to MDWs?”, (2) ”what are the functions/outcomes of peer support for MDWs?”, and (3) “what are the barriers and facilitators for MDWs to provide/receive peer support?”.

### 2.3. Identifying Relevant Information

According to Peters et al. (2020), keywords were identified by an initial literature search using very board relevant terms (e.g., peer, migrant domestic workers, mental) on two online databases (CINAHL and OVID PsycInfo) [42]. An analysis of the text words contained in the title and abstract of retrieved papers, and of the index used to describe the articles was then conducted. Studies published between January 2001 and October 2021 were then identified through a systematic search of eight electronic databases, including OVID Medline, OVID PsycInfo, EBSCO CINAHL, EBSCO Business Source Complete, Scopus, Web of Science Core Collection, ProQuest, and PubMed. We also hand-searched social media platforms (e.g., Google search, LinkedIn) and research registers (US National Institutes of Health Clinical Trials Register and WHO International Clinical Trials Registry Platform). In addition, references from the included studies were also checked for additional relevant articles. Search terms for this review included “domestic helper”, “domestic worker”, “migrant worker”, “peer group”, “peer*”, and “mutual support”. The search strategies used in OVID Medline are presented in Appendix A.

### 2.4. Study Selection

The study selection was guided by the Population, Concept, and Context framework [42]. The inclusion and exclusion criteria for this review are listed in Table 1. All search records retrieved by electronic searching were imported to Mendeley Desktop and duplicates were removed. All remaining citations were exported to the Covidence review manager for review and screening. Two independent reviewers (C.Y. and A.K.Y.L.) screened the titles and abstracts and then full-text articles to decide whether they met the inclusion/exclusion criteria. Disagreements between the two reviewers were resolved by discussion, with the involvement of a third researcher (K.H.M.H), whenever necessary. Potentially relevant studies were initially identified from screening the titles and abstracts. The full texts of those found relevant were assessed for eligibility. We also manually searched the reference lists of eligible reports and articles to identify any additional relevant studies that may have been missed.

### 2.5. Charting the Data

The research team developed the data charting form. The data items collected included: general information (title, authors, year, country, and study design), study population and its characteristics, types of peer support, functions/outcomes of peer support, and facilitators of and barriers to peer support. Two reviewers (C.Y. and A.K.Y.L.) extracted data from the included studies independently, and disagreements were resolved through discussion or consulting the third researcher (K.H.M.H.).

### 2.6. Collating, Summarizing, and Reporting the Results

Considering this review included experimental and nonexperimental study designs, results were integrated through narrative synthesis. Content analysis was performed under the framework of the three main research questions of this review [43]. Results were presented in thematic narratives, as suggested by Levac et al. [41].

## 3. Results

The literature search identified 3345 articles. After removing duplicate records, 1702 articles were eligible for the title and abstract screening. Among these articles, 1680 were excluded after screening the titles and abstracts, leaving 22 articles for the full-text screening. Thirteen articles were further excluded with the remaining nine articles included in this review. Moreover, another three articles were identified through checking the references of the included studies and the Google search engine. Finally, 12 articles were included for the review. Figure 1 shows the PRISMA flow diagram of article searching and screening in this review.

### 3.1. Characteristics of Included Studies

Two of the 12 included articles had a quantitative study design [44,45], four had a qualitative study design [46,47,48,49], and five had a mixed-methods study design [24,50,51,52,53]. One article was a news report [54]. Four of the articles were conducted in Hong Kong SAR, China [45,48,49,50]; three in Singapore [51,53,54]; two in Macao SAR, China [44,47]; another three from each of the three countries, including South Korea [52], Philippines [24], and Canada [46]. The number of participants of the included articles varied significantly from 5 [52] to 2017 [50], and the mean or median age ranged from 35.1 [24] to 42.9 years [47]. The studied populations included Filipino (*n* = 6), Indonesian (*n* = 1) [49], and a mixed sample of Filipino and Indonesian (*n* = 3) [45,48,50]. All articles, except one [50], reported the percentage of female participants. One article included 93.9% female participants [46], while the remaining 10 articles had 100% female. Table 2 presents the detailed characteristics of all included studies.

### 3.2. What Types of Peer Support Are Available to MDWs?

Two types of peer support were identified according to the roles played by participants, i.e., mutual aid and para-professional trained peer support.

#### 3.2.1. Mutual Aids

Nine articles focused on addressing MDWs’ mental health problems via mutual aid [24,44,45,46,47,48,49,50,52]. Mutual aid refers to the reciprocal support and help between peers with similar experiences [46]. Mutual aid was mostly rooted in the reciprocity between people with the relevant lived experience of the concerned problems, without formal training or involvement of healthcare professionals [55]. The structural marginalization of MDWs from the health and social care system leaves MDWs with mutual aid as their only option [46,47]. In Canada, the social care services usually exclude temporary foreign workers and are only available during the normal working days when MDWs are fulfilling their work duties [46]. Other issues included poor knowledge of the healthcare system in host countries, communication problems because of language, and cultural isolation [47]. All of these factors made MDWs tend to seek informal support from their peers. As such, MDWs were eager to develop friendship networks and build trust among MDWs in host countries [46]. A majority of MDWs gave/received mutual aid to/from their informal social networks, particularly from their friends or fellow workers in host countries [24,46]. For example, the majority of MDWs in Singapore preferred to seek help from peers (43%) rather than from professionals (2%) when faced with emotional problems [51]. Another survey in Hong Kong also found that over half of MDWs reported daily contacts with their local friends [45].

#### 3.2.2. Para-Professional Trained Peer Support

Another type of peer support was para-professional trained peer support, which trained MDWs as peer supporters with some professional helping or counselling skills to provide peer support for other MDWs with supervision and support from healthcare professionals, such as psychologists [51,53,54]. Wong and colleagues [51,53,54] developed a mental health para-professional training program for Filipino MDWs in Singapore. The program aimed to equip MDWs with skills to recognize depressive symptoms and utilize Cognitive Behavioral Therapy (CBT) skills to support their peers.

### 3.3. What Are the Functions/Outcomes of Peer Support for MDWs?

#### 3.3.1. Mutual Aids

MDWs mainly sought support from peers through mutual aid for emotional support [24,45,48,49,50,52], informational support [46,49,50,52], and instrumental support [49,50,52]. A large survey of 2017 MDWs in Hong Kong showed that 57.4% of Filipino MDWs and 42.8% of Indonesian MDWs sought emotional support from their friends in Hong Kong, while only 5.1% and 6.5% of them found emotional support from social workers and NGOs, respectively [50]. Three articles found that mutual aid was one of the most important strategies for MDWs to reduce their stress and burden [24,45,52]. It could also offer a source for MDWs in psychological distress to relieve traumatic experiences related to work and help them reconnect with ordinary life [48]. Peer support relationships were mutual and reciprocal, as each of the MDWs involved could obtain potential benefits to their mental well-being [52]. Moreover, compared with formal supporting systems that focused more on problem-solving, peer support might provide emotional comfort nonjudgmentally and was considered to be better placed to address mental health problems of these MDWs [50].

Further, MDWs could also obtain informational and instrumental support from their peers, such as shelters [50], job offers [52], and protective gear, during the COVID-19 pandemic [49]. This support provided emotional comfort and improved the mental well-being of MDWs. In addition, peer support provided a source that connected MDWs to the formal mental healthcare system [50].

#### 3.3.2. Para-Professional Trained Peer Support

Results of a piloted trial of the para-professional training program showed significant positive effects on improvements of depression literacy, CBT knowledge, and the stigma associated with depression in trained MDW peer supporters after the intervention [51]. At the 2-month follow-up, trained MDW peer supporters had a significantly improved depression literacy and attitudes toward seeking professional psychological support [51]. However, no significant change was observed in terms of MDW peer supporters’ confidence in supporting people with depression after the training. It is encouraging that more than 70% of the trained MDW peer supporters indicated that they were willing to be peer counselors [53].

### 3.4. What Are the Barriers and Facilitators for MDWs to Provide/Receive Peer Support?

#### 3.4.1. Facilitators

##### Mutual Aids

The collective culture of Filipino and Indonesian MDWs was identified to be a facilitator of peer support. In Filipino culture, the “bayanihan” is the foremost social value and refers to the spirit of helping community members in times of need without expecting anything in return [46]. For Indonesians, women are not encouraged to work outside; and family conflict can sometimes be a source of emotional distress. Therefore, they are more willing to seek help from their friends [50]. One study also placed emphasis on the role of religion among MDWs, which formed a major source for them to establish friendship through the social networks of religious organizations [24]. In addition to face-to-face communication, MDWs also gained mutual aid via mobile phone chat groups [52] and social media (e.g., Facebook) [46,49,52].

##### Para-Professional Trained Peer Support

The interventional study of the para-professional trained peer support program identified several facilitators of peer supporters to provide peer support [51,53]. Most of the trained peer supporters preferred providing the peer support service in a mixture of their native language and English. They were more willing to offer services through face-to-face contacts, instead of using telephone, email, text message, and/or social media [53]. Continued education and supervision by mental health professionals were suggested to improve their skills and confidence to provide these services [51,53].

#### 3.4.2. Barriers

##### Mutual Aids

A particular phenomenon mentioned in three articles [44,47,52], that is, “contagion of emotion or stress”, occurred when one’s negative emotions were transferred to others in peer communication, leading others to have similar emotions without their awareness. In addition, concerns about the spreading of personal information and happenings around the MDW community might lead to distrust and increase MDWs’ stress [44,47,50]. Due to the self-help nature of mutual aid, peers provided support to others mostly on the basis of their personal experiences, which were not always helpful in different situations [24,52]. Some MDWs could not receive favorable feedback from their peers, or even became more stressed due to the exposure to peer judgment and stigma [46,49]. Employers could also be a barrier to MDWs’ receiving peer support. A study in Hong Kong reported that some exploitive employers locked MDWs inside their apartment/house and did not allow them to socialize with their peers [48]. The same study also revealed that employers even confiscated the passports of MDWs to purposively isolate them. As such, these employers made sure that MDWs were unable to socialize with other MDWs, as a means of control and dehumanization [48].

##### Para-Professional Trained Peer Support

Barriers to peer supporters offering peer support include their unavailability during working days, lack of support from culturally competent health and social care professionals, and the varied length-of-stay period of MDWs in host countries [51,53]. In addition, the overwhelming nature of the newly learnt knowledge and skills and tight schedule of training lowered their self-efficacy to provide peer support [51,53].

## 4. Discussion

This was the first scoping review that synthesized evidence on peer support among MDWs and explored the facilitators of and barriers to peer support in this vulnerable and marginalized population. Our review highlighted the importance of peer support among MDWs that relieved their psychological distress and improved their mental well-being, providing a valuable addition to current literature about mental health care and support for MDWs.

The findings of the review suggests that mutual aid is the most popular type of peer support among MDWs, as indicated in nine of the 12 included articles. Compared with formal mental healthcare system, mostly MDWs preferred seeking emotional comfort from their informal social networks. Their mental health needs were not met by existing formal support services in the host countries. This finding is consistent with a cross-sectional survey, in which 43% of MDWs in Singapore asked for help from peers when faced with emotional problems, while only 2% of them sought help from professionals [56]. Most of the included studies indicated that MDWs obtained emotional support and a sense of belonging from their peers or friends who shared common issues and had similar cultural backgrounds. The information sharing and instrumental support gained from peer support also allowed for emotional comfort and eased their psychological distress. Most importantly, providing mutual aid and assistance was culturally appropriate and relevant for MDWs, such as the concept of “bayanihan” in Filipino culture [46]. Therefore, mutual aid groups could be a complement to the formal mental healthcare system/service.

Although mutual aid groups could bring about positive changes in MDWs’ psychological distress, some associated psychological problems were also noted. First, negative emotion could be spread between an individual and a peer, causing a “contagion of emotions”. The observation of another person’s emotional state would automatically activate the same nervous system response as that of the observer, and awaken affective arousal and contagion [57]. This process might potentially cause harm to others. In addition, peers in mutual aid were often in lack of appropriate or effective strategies to handle mental health problems. Finally, peers could also be a source of stress, given that the worries about the disclosure of their personal information and the spread of rumors were frequently mentioned, leading to distrust and conflict among the MDW community [44,46,47,49,50,52]. Therefore, mutual support might cause negative emotion arousal and was even associated with additional distress. It has been suggested that the involvement of trained professional workers is essential to overcome the challenges in providing mutual support [58]. There is a clear need for MDW peer supporters to receive training to handle interpersonal relationships and issues from mental health professionals to learn effective psychosocial support and communication skills before moving to an eligible peer support worker. No structured, professional-support mutual aid group was identified among MDWs from the included studies, suggesting an important service gap in this field. Furthermore, the identified para-professional trained peer support program only focused on equipping MDWs with CBT skills and knowledge, without covering topics of interpersonal relationships and issues (e.g., contagion of emotions). The lack of training in handling interpersonal relationships and issues, particularly within the scope of peer support, presents a prominent gap for a peer support program to fill up in the future.

The para-professional trained peer support appeared to be a feasible way to maximize the expertise of peer support workers and improve the mental health knowledge and skills of trained MDW peer supporters. After receiving training from mental health professionals, trained MDW peer supporters had significant improvements in professional helping and supporting skills, such as CBT knowledge and depression literacy, thereby enhancing their ability to recognize and tackle their peers’ mental health problems [51]. The training program also improved their attitudes toward seeking professional support and assistance by acting as a bridge or mediator between the MDW community and formal mental healthcare system. It was encouraging that the majority of trained MDWs were willing to use their expertise to assist others as a peer counselor, indicating that the para-professional trained peer support could be feasible among MDWs as a component of current mental healthcare services [53]. However, this study [51] is a feasibility trial involving only 40 Filipino participants, limiting its ability to detect reliable estimates of intervention effects. Furthermore, the trained MDW peer supporters did not start their service to MDWs in distress. Therefore, the health outcomes of those supported by trained peer supporters were unknown. A larger-scale controlled study is needed to evidence the effectiveness of peer support in wider MDW samples in different ethnic and socio-demographic contexts. There were also some barriers to the successful implementation of the para-professional trained peer support found in this review, such as the limited hours of work on weekdays; the lack of ongoing supervision, training, and sufficient resources; the lack of culturally competent health and social care professionals [51,53]. Similar challenges in serving as a peer support worker for promoting mental health were also reported in review and qualitative studies [59,60,61]. More efforts should be directed toward selecting appropriate strategies to overcome these barriers. This may need close cooperation between MDWs, employers, nongovernment healthcare organizations, mental healthcare providers, and other stakeholders (e.g., consulates of migrant countries).

### 4.1. Limitations

This review had several limitations. Although an extensive search of the literature and grey literature was conducted, we only retrieved articles written in English and did not have access to other languages. Given that a large number of MDWs are working in Western Asia and the Middle East [6], studies published in the Arabic language can be included in a future review. We did not assess the quality of each of the included studies, due to the nature of a scoping review. For instance, one article included was a news report [54]. However, this is also a strength of a scoping review to include grey literature for inductive purposes.

Particularly, all the included studies were from Asia, except one from Canada. Caution should be taken when generalizing the study’s findings to Western countries. Meanwhile, it is noteworthy that only Canada provides opportunities for MDWs to be permanent residents. The particular context of the to-be-permanent residency of Canada may be a concern not experienced by other MDWs in Asia or Europe. Yet, our scoping review identified very limited studies from Western countries, presenting a research gap here.

Another limitation is the over-representation by female MDWs in this scoping review. Although it is well-acknowledged that over 70% of MDWs are female, the under-representation of male MDWs in this review may uncover the particular vulnerability of male MDWs: being invisible in a marginalized and vulnerable population. Our findings may not be generalizable to male MDWs. As such, there is an urgent need to explore the problems faced by male MDWs worldwide.

A final point to note is that while employers are influential on the wellbeing of MDWs [45], we were unable to provide examples of how employers facilitate peer support for MDWs from identified studies. Instead, our findings showed that employers could provide a barrier to peer support, in accordance with other studies (e.g., Van der Ham) [62]. Future studies of how employers influence the provision and receipt of peer support for MDWs are warranted.

### 4.2. Implications for Research and Practice

Our review shows that training MDWs as para-professional peer supporters is feasible and may be potentially effective to improve the mental health knowledge and skills of trained MDWs. The engagement of MDWs and service providers may assist in developing culture-specific interventions and overcoming barriers to peer support. Yet, empirical evidence from randomized controlled trials on the effectiveness of mutual aid or para-professional peer support on the mental health outcomes of those being supported was unavailable. Future trials with a robust design and methodology are required to identify the most effective peer support strategies or programs for MDWs in mental health promotion.

Peer support is an important complement to the traditional mental support system. Mental healthcare services and relevant government and nongovernment organizations should establish structured peer support services, either in mutual self-help groups or professional-led peer support, to make them available to MDWs in need of such support. Healthcare professionals, together with the mental healthcare organizations and employment agents, should work hand-in-hand to provide training programs for the MDW peer leaders. Meanwhile, the government of the host countries may require employment contracts to provide stipulated study leave for MDWs to receive regular training in psychological care skills and knowledge for peer support.

## 5. Conclusions

Instead of formal mental healthcare systems, MDWs mainly seek emotional support from peers in their host countries. Para-professional trained peer support has shown some benefits for improving the mental health knowledge and skills of trained MDW peer supporters. The facilitators of and barriers to peer support among MDWs are identified from current best available evidence, and which may serve as the basis for a future work or study in designing and testing a peer support intervention for improving the mental health of MDWs in different countries. Yet, empirical evidence of both mutual aid groups and para-professional trained support on the mental health outcomes of MDWs being supported was unavailable. Large-scale controlled trials with a robust study design are suggested to examine the effects of peer support interventions for MDWs with diverse ethnic and cultural backgrounds.

## Figures and Tables

**Figure 1 ijerph-19-07617-f001:**
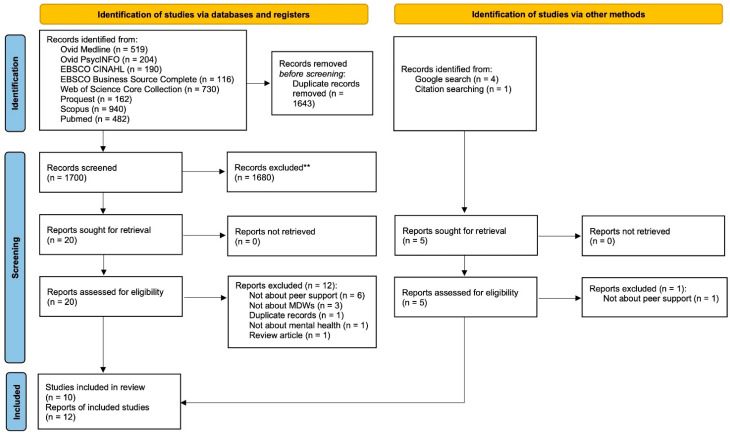
PRISMA flow diagram. ** upon title and abstract screening.

**Table 1 ijerph-19-07617-t001:** Inclusion and exclusion criteria for this review.

	Inclusion Criteria	Exclusion Criteria
Population	Migrant domestic workers (MDWs) aged 18 years and over	Migrant workers not engaged in domestic work
		A mixed international migrant and native workers group without a separate description of study outcomes related to MDWs
Concept	Interventions/services/phenomenon that are peer-support-focused and aim to address MDWs’ mental health problems	Studies that do not list peer support as a core or critical element of the intervention/service/phenomenon
Context	Any setting, such as the home, hospitals, clinics, and community organizations	-
Study design	Quantitative study, qualitative study, and mixed-methods study	Conference abstracts, reviews, editorials, or commentary articles

**Table 2 ijerph-19-07617-t002:** Characteristics of included studies.

Author, Year, Country	Aim of Study	Study Design	Sample Characteristics	Type of Peer Support	Functions/Outcomes of Peer Support and Mental Health	Facilitators for	Barriers for
						Peer Support	Peer Support
Baig and Chang, 2020 [50] Hong Kong SAR, China	To explore how migrant domestic workers (MDWs) approach different forms of support systems based on their multiple identities of gender, ethnicity, and religion.	Mixed-methods study	Sample size: quantitative (*n* = 2017); qualitative (*n* = 18) Age mean (years): Not reported Female%: Not reported Nationality: Filipino and Indonesian	Mutual aid	A significantly higher proportion of Filipino MDWs sought emotional support from friends than Indonesian MDWs (*p* ≤ 0.01). A significantly higher proportion of Filipino MDWs sought health-related support from friends compared with Indonesian MDWs (*p* ≤ 0.01). Peers provide emotional support, informational support, and instrumental support. Peers can be the sources that connect MDWs to formal support systems.	NA	Worries about personal information and happenings spreading around the MDW community.
Bhuyan et al., 2018 [46] Canada	To explore MDWs’ response to employer abuse and exploitation following changes to Live-in-Caregiver Program in 2014.	Qualitative study	Sample size: 33 Age median (years): 38.6 Female%: 93.9 Nationality: most are Filipino (*n* = 31)	Mutual aid	Sharing information and experiences with peers. Forming and maintaining relationships and friendships. Proving a sense of community.	The “bayanihan” in Filipino culture encourages individuals to help each other.	Exposure to peer judgement, stigma, and insecurity.
Hall et al., 2019 [47] Macao SAR, China	To identify key health issues Filipino MDWs were facing in their post-migration context, and the social determinants of these issues.	Qualitative study	Sample size: 22 Age mean (years): 42.9 Female%: 100 Nationality: Filipino	Mutual aid	NA	NA	Rumors spreading among MDW community. Being with peers who are similarly stressed and lacking control may not be good for mental health.
Ladegaard, 2015 [48] Hong Kong SAR, China	To investigate how the women make sense of their traumatic experiences, and how peer support becomes essential in the narrators’ attempts to rewrite their life stories from victimhood to survival and beyond.	Qualitative study	Sample size: 41 Age mean (years): Not reported Female%: 100 Nationality: Indonesian and Filipino	Mutual aid	MDWs needed emotional support.	NA	Some employers may lock MDWs inside the flat and/or confiscate their passports, making sure that MDWs do not get to socialize with other workers.
Mendoza et al., 2017 [44] Macao SAR, China	To determine the role of social network support in buffering the impact of post-migration stress on mental health symptoms among Filipino MDWs.	Quantitative study	Sample size: 261 Age mean (years): Not reported Female%: 100 Nationality: Filipino	Mutual aid	Social network support from friends was positively associated with depressive symptom severity (*p* < 0.001), anxiety symptom severity (*p* < 0.001), and post-traumatic stress disorder symptom severity (*p* < 0.05). Social network support from friends was not associated with somatization (*p* = 0.06).	NA	Gossip and rumors that circulate in MDWs’ social networks may lead to distrust and conflict. Contagion of emotion or stress.
Oktavianus and Lin, 2021 [49] Hong Kong SAR, China	To explore how the storytelling networks of MDWs provided social support amid the COVID-19 pandemic.	Qualitative study	Sample size: 32 Age mean (years): 37.7 Female%: 100 Nationality: Indonesian	Mutual aid	Peers may provide informational assistance, emotional comfort, and instrumental support.	NA	May receive unfavorable feedback from their peers. Exposure to fake news from interpersonal networks.
van der Ham et al., 2014 [24] Philippines	To provide insight into the resilience of female domestic workers by presenting the results of an exploratory study on resilience in which personal resources and social resources were investigated in relation to perceived stress and well-being.	Mixed-methods study	Sample size: quantitate (*n* = 500); qualitative (*n* = 21) Age mean (years): 35.1 Female%: 100 Nationality: Filipino	Mutual aid	Peers provide emotional support. Participants who used “talking to a friend” to deal with stress as a coping strategy had a significantly higher well-being than those who did not use this coping strategy (*p* < 0.05). The self-perceived stress level was similar for participants who use “talking to a friend” to deal with stress and those who did not (*p* > 0.05).	Being a member of a religious organization.	Endurance and acceptance as coping strategies.
Wong et al., 2020; Suthendran et al., 2017; Hui, 2016 [51,53,54] Singapore	To assess the acceptability and effectiveness of a Cognitive-Behavioral-Therapy-based para-professional training program for Filipina MDWs.	Mixed-methods study	Sample size: 40 Age mean (years): 38.6 Female%: 100 Nationality: Filipino	Para-professional trained peer support	Following training, the intervention group and wait-list group showed improved depression literacy (*p* = 0.002; Cohen’s dz = 0.55) and CBT knowledge (*p* = 0.02; Cohen’s dz = 0.42) and decreased stigma associated with depression (*p* = 0.03; Cohen’s dz = −0.36). There was no increase in confidence in supporting individuals with depression and attitude toward seeking professional psychological support (*p* > 0.05). At the two-month follow-up, the intervention group and wait-list group showed improved depression literacy (*p* = 0.001; Cohen’s dz = 0.59) and attitude toward seeking professional psychological support (*p* = 0.03; Cohen’s dz = 0.38). There was no increase in other outcomes (*p* > 0.05). No difference was found on changes in any outcome variables in the intervention group as compared to the wait-list group (*p* > 0.05).	Training using a mixture of local language and English. Training offered by social media and texting. Continuing education and training. Supervision of peer counsellors by professionals.	Availability to be peer counsellors: most were only available on Sundays, with a minority available on weekday evenings. Difficulties utilizing CBT skills. Difficulties in recruitment of peer counsellors: the transient nature of MDWs poses a structural service dilemma. Lack of culturally competent health and social care professionals.
Wrigglesworth, 2016 [52] South Korea	To describe the mobile phone ego networks of Filipino MDWs living in Korea; To understand what these mobile phone ego networks mean to MDWs.	Mixed-methods study	Sample size: 5 Age mean (years): 37.4 Female%: 100 Nationality: Filipino	Mutual aid	Peers provide informational support, emotional support, and instrumental support.	The use of mobile phones allows MDWs to stay in contact with others while on the move.	Cultural barriers preventing true understanding of the problem. Contagion of emotion or stress.
Ye and Chen, 2020 [45] Hong Kong SAR, China	To explore the potential health buffering effects of MDWs’ personal network.	Quantitative study	Sample size: 1695 Age (years): 50% between 30 and 40 Female%: 100 Nationality: Indonesian and Filipino	Mutual aid	MDWs needed emotional support. “Contact friends in Hong Kong daily” (*p* < 0.01) and “often come to gathering with foreign domestic workers” (*p* < 0.01) were significantly associated with good self-reported health. “Contact friends in other countries daily” was not significantly associated with self-reported health (*p* > 0.05).	NA	Religious activity participation may not be an effective coping strategy.

*Note*. CBT = Cognitive Behavioral Therapy; MDW = Migrant Domestic Worker; NA = Not Applicable.

## Data Availability

No new data were created from or analyzed in this study.

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
