# Peer review of "Peer Support and Mental Health of Migrant Domestic Workers: A Scoping Review"

_ijerph, 2022, doi:10.3390/ijerph19137617_

Round 1
Reviewer 1 Report
This manuscript is an excellent comprehensive assessment of the literature that indicates a significant knowledge gap regarding migrant domestic workers and the facilitation of support. A scoping review aimed to identify the types of available evidence available. The significance of this publication stems from its complete presentation of the research available to advise readers and clinicians on the subject. This manuscript is generally well-written and well-researched. It is illuminating and necessary for our comprehension.
Minor revisions to consider for authors:
Lines 74–75: Authors might add more information here. It is commonly acknowledged that peer assistance within collectivist societies is critical for coping and mental health support.
Lines 110 – 113: The gap needs to be communicated better. The research's need is clearly within the manuscript's scope; however, more emphasis is needed on the purpose of the scoping review.
Lines 134-136: What was the process of identifying the key search terms to ensure broad coverage of available literature?
Line 140: Were full texts obtained or only used abstracts to identify studies? The full text could give an additional possibility of unidentified literature to use within the scoping review.
Line 187: I would suggest subheading findings with research questions for structure and readability.
Line 190: Capitalize the ‘mutual.'
Line 280: Employers are briefly listed here. When employers play a significant part in this literature, I would anticipate them to have a significant influence on peer support. More detail to back this up could provide a more complete story and analysis. To what extent do employers prevent/facilitate support for a migrant domestic worker. Is this applicable to all employers? If not, what is the reason?
Line 365: Although the authors have mentioned the overrepresentation of female migrant workers in the literature and findings, it would be beneficial to discuss this in the limits or discussion. Consider the gender influence or a critique of the research's generalisability. A critique of the lack of research into male migrant domestic workers could be picked up within this scoping review. Furthermore, have the authors identified any other gaps within the studies used in the scoping review? The literature of the studies included could be delved into deeper to identify the gaps to address the knowledge surrounding peer support.
Author Response
Dear Editor and Reviewers,
Thank you very much for the positive and helpful comments on our manuscript. The suggested revisions are indeed helpful for us to improve the paper before publication. We have revised the manuscript in response to your comments one-by-one at below.
Reviewer 1
Minor revisions to consider for authors:
Comment 1: Lines 74–75: Authors might add more information here. It is commonly acknowledged that peer assistance within collectivist societies is critical for coping and mental health support.
Response: The requested information is added accordingly with reference. (Line 85-86)
Reference: Fisher, E. B., Coufal, M. M., Parada, H., Robinetter, J. B., Tang, P. Y., Urlaub, D. M., Castillo, C., Guzman-Corrales, L. M., Hino, S., Hunter, J., Katz, A. W., Symes, Y. R., Worley, H., & Xu, C. (2014). Peer support in health care and prevention: Cultural, organizational, and dissemination issues. Annu. Rev. Public Health, 35, 363-83. Doi: 10.1146/annurev-publhealth-032013-182450
Comment 2: Lines 110 – 113: The gap needs to be communicated better. The research's need is clearly within the manuscript's scope; however, more emphasis is needed on the purpose of the scoping review.
Response: The gap is further strengthened by adding “there is a prominent need to better understand about the importance and the relevant factors influencing peer support before its use in MDWS to promote the mental health of MDWs” (Line 122-124).
Comment 3: Lines 134-136: What was the process of identifying the key search terms to ensure broad coverage of available literature?
Response: According to Peters et al. (2020), keywords were identified by an initial literature search using very board relevant terms (e.g., peer, migrant domestic workers, mental) on two online databases (CINAHL and OVID PsycInfo). An analysis of the text words contained in the title and abstract of retrieved papers, and of the index used to describe the articles was then conducted.(Line 141 – 145)
Reference:
Peters, M.D.J.; Godfrey, C.; McInerney, P.; Munn, Z.; Tricco, A.C.; Khalil, H. Chapter 11: Scoping Reviews (2020 Version). In JBI Manual for Evidence Synthesis, JBI, 2020.; Aromataris, E., Munn, Z., Eds.; 2020.
Comment 4: Line 140: Were full texts obtained or only used abstracts to identify studies? The full text could give an additional possibility of unidentified literature to use within the scoping review.
Response: Potentially relevant studies were initially identified from screening the titles and abstracts. The full texts of those found relevant were assessed for eligibility. We also manually searched the reference lists of eligible reports and articles to identify any additional relevant studies that may have been missed. (Line 164-167)
Comment 5: Line 187: I would suggest subheading findings with research questions for structure and readability.
Response: Subheadings are changed as suggested.
Comment 6: Line 190: Capitalize the ‘mutual.'
Response: Done.
Comment 7: Line 280: Employers are briefly listed here. When employers play a significant part in this literature, I would anticipate them to have a significant influence on peer support. More detail to back this up could provide a more complete story and analysis. To what extent do employers prevent/facilitate support for a migrant domestic worker. Is this applicable to all employers? If not, what is the reason?
Response: We have added some additional discussion about how employers may be barriers for the implementation of peer support. For example, lines 304-307 now state:
The same study also revealed that employers even confiscated the passports of MDWs to purposively isolate them. As such, these employers made sure that MDWs were unable to socialise with other MDWs, as a means of control and dehumanization.
However, among the identified articles, there were not information about how employers facilitate social/peer support for a migrant domestic workers. In fact, employers were more likely to be a source of social negativity rather than social support, since employers’ abuse and restrictions significantly intensify workers’ stress. Therefore, we also strengthened the limitation of study as below (line 407-412):
A final point to note is that while employers were suggested to be influential on the wellbeing of MDWs, we were unable to provide examples of how employers facilitate peer support of MDWs from identified studies. Instead, our findings showed that employers could be a source of barrier of peer support in accord with other studies (e.g. Van der Ham). Future studies of how employers influence the provision and receipt of peer support of MDWs is warranted.
Reference:
Van der Ham, A. J., Ujano-Batangan, M. T. Ignacio, R., & Wolffers, I. (2015). The dynamics of migration-related stress and coping of female domestic workers from the Philippines: An exploratory study. Community Mental Health Journal, 51(1), 14-20. https://doi.org/10.1007/s10597-014-9777-9
Comment 8: Line 365: Although the authors have mentioned the overrepresentation of female migrant workers in the literature and findings, it would be beneficial to discuss this in the limits or discussion. Consider the gender influence or a critique of the research's generalisability. A critique of the lack of research into male migrant domestic workers could be picked up within this scoping review. Furthermore, have the authors identified any other gaps within the studies used in the scoping review? The literature of the studies included could be delved into deeper to identify the gaps to address the knowledge surrounding peer support.
Response: We added some discussion about the gap regarding the lack of training on interpersonal relationships in peer support programmes and the lack of information about male MDWs (Line 349-358; Line 401-406).
Line 349-358:
There is a clear need for MDW peer supporters to receive training to handle interpersonal relationships and issues from mental health professionals to learn effective psychosocial support and communication skills before moving to an eligible peer support worker. No structured, professional-support mutual aid group was identified among MDWs from the included studies, suggesting an important service gap in this field. Furthermore, the identified para-professional trained peer support programme only focused on equipping MDWs with CBT skills and knowledge, without covering topics of interpersonal relationships and issues (e.g., contagion of emotions). The lack of training in handling interpersonal relationships and issues, particularly within the scope of peer support, presents a prominent gap for peer support programme to fill up in the future.
Line 401-406:
Another limitation is the over-representation by female MDWs in this scoping review. Although it is well-acknowledged that over 70% of MDWs are female, the under-representation of male MDWs in this review may uncover the particular vulnerability of male MDWs: being invisible in a marginalized and vulnerable population. Our findings may not be generalizable to male MDWs. As such, there is an urgent need to explore the problems faced by male MDWs worldwide.
Reviewer 2 Report
This is a very important contribution to the field of understanding the problems and challenges surrounding the migrant domestic workers throughout the world, and more particularly in those areas such as the Middle-Eastern and Asian geographical space.
One issue I found with the paper is that it focuses mainly, obviously, on Asian countries relying on MDWs. There might be clear reasons why this is the case, but it seems to me important to offer some sort of comparison to Western nations and how they handle this phenomenon - or to open the discussion for further research.
Author Response
Dear Editor and Reviewers,
Thank you very much for the positive and helpful comments on our manuscript. The suggested revisions are indeed helpful for us to improve the paper before publication. We have revised the manuscript in response to your comments one-by-one at below.
Comment 1: One issue I found with the paper is that it focuses mainly, obviously, on Asian countries relying on MDWs. There might be clear reasons why this is the case, but it seems to me important to offer some sort of comparison to Western nations and how they handle this phenomenon - or to open the discussion for further research.
Response: The scoping review was only able to identify one study from a Western nation (Canada) and most studies for MDWs were conducted in Asian countries. Meanwhile, Canada is the only nation allowing MDWs to be permanent residents. This context was supplemented in the discussion. (Line 397-400)
Line: 397 – 400:
Particularly, all the included studies were from Asia, except one from Canada. Cautions should be taken when generalizing the study findings to western countries. Meanwhile, it is noteworthy that only Canada provides opportunities for MDWs to be permanent residents. The particular context of to-be-permanent residency of Canada may be a concern not experienced by other MDWs in Asia or Europe. Yet, our scoping review identified very limited studies from Western countries, presenting a research gap here.
Reviewer 3 Report
Introduction: a more descriptive information about the tasks and job specification might be helpful. For example, are the MDWs only Aupair workers? Do they need any kind of proficiency, e.g. language or technical? How long do they usually stay in the host countries?
Regarding the prevalence of depression (10.3 - 18.2%, Line 53). How is this prevalence in other work groups, e.g. cleaning personal? Is this prevalence very different from what we find in the population?
How is the distribution for MDWs, that are at risk for high psychological distress, outside Hong Kong?
It is known from the literature, that stress negatively impacts quality of life, not only for MDWs (Lines 57-58). Are only the MDWs being overlooked by the healthcare system? (lines 64-65) or is this a systematic problem? Methods: why did the authors not perform a meta analysis on this dataset?
Table 2:
it would be very informative to add effect sizes to the table.
Why not run a meta analysis?
Please correct the p-values: p<= 0.01. Are the values smaller (<) or equal (=) to 0.01? The p-values reported as p< 0.05 should also be entered correctly, please report the exact value, e.g., p = 0.025. Usually for such cases: p < 0.0001 the p-value should be reported this way (smaller than a specific value). The same for p-value > 0.05. It would be very informative to have the actual values.
Author Response
Thank you very much for the positive and helpful comments on our manuscript. The suggested revisions are indeed helpful for us to improve the paper before publication. We have revised the manuscript in response to your comments one-by-one at below.
Comment 1: Introduction: a more descriptive information about the tasks and job specification might be helpful. For example, are the MDWs only Aupair workers? Do they need any kind of proficiency, e.g. language or technical? How long do they usually stay in the host countries?
Response: We have clarified that internationally, there are not universally agreed job specifications or proficiency requirements for MDWs. Generally, MDWs perform cooking, cleaning and caring for children or older adults (Line 37-40). Their length of stay also varies across countries, depending on the length of employment contract (e.g., 2 years in Hong Kong) or rights to permanent residency (e.g., Canada). (Line 45-48)
Comment 2: Regarding the prevalence of depression (10.3 - 18.2%, Line 53). How is this prevalence in other work groups, e.g. cleaning personal? Is this prevalence very different from what we find in the population?
Response: We were unable to identify a comparable job (e.g. cleaning personal) to compare the prevalence of depression of MDWs. Instead, 19% of general population in Hong Kong had depression during the COVID-19 pandemic. As such the prevalence of 10.3-18.2% of MDWs suffered from moderate to severe levels of depression across countries during non-COVID time was comparable to the general population. Yet, a recent cross sectional study showed that 40% of MDWs were at high risk of depression and anxiety. Revision is as below (line 58-62):
Estimates from a recent review show that 10.3¬–18.2% of MDWs suffered from moderate to severe levels of depression across countries [10], which was comparable to that of general population in Hong Kong (19% had depression as measured by the patient health questionnaire-9) during the COVID-19 pandemic [17]. A cross-sectional study involving 346 MDWs in Hong Kong found that near 40% of them were at risk for high psychological distress, as measured by the 14-item Hospital Anxiety and Depression Scale [15].
References:
Choi, E. P. H., Hui, B. P. H., & Wan, E. Y. F. (2020). Depression and anxiety in Hong Kong during COVID-19. International Journal of Environmental Research and Public Health, 17, 3740. Doi: 10.3390/ijerph17103740
Comment 3: How is the distribution for MDWs, that are at risk for high psychological distress, outside Hong Kong?
Response: A survey in Taiwan showed that 16.5% of Indonesian MDWs had depressive symptoms (Palupi et al., 2017). A study in Macao including 1,375 Filipino MDWs found that 38% and 32% of them met the criteria for depression and anxiety, respectively (Garabiles et al. 2019). (Line 66-69)
References
Garabiles, M. R., Lao, C. K., Xiong, Y., & Hall, B. J. (2019). Exploring comorbidity between anxiety and depression among migrant Filipino domestic workers: A network approach. Journal of affective disorders, 250, 85–93.
Palupi, K. C., Shih, C. K., & Chang, J. S. (2017). Cooking methods and depressive symptoms are joint risk factors for fatigue among migrant Indonesian women working domestically in Taiwan. Asia Pacific journal of clinical nutrition, 26(Suppl 1), S61–S67. https://doi.org/10.6133/apjcn.062017.s3
Comment 4: It is known from the literature, that stress negatively impacts quality of life, not only for MDWs (Lines 57-58). Are only the MDWs being overlooked by the healthcare system? (lines 64-65) or is this a systematic problem?
Response: Thanks for your comments. MDWs are one of the most vulnerable populations who are often overlooked and underrepresented in the healthcare system. According to the International Labor Organization (ILO, 2013), MDWs are usually not protected under labor laws of host countries. They had inadequate health insurance for medical diseases in host countries compared to local populations. The lack of consideration of their cultural and linguistic variation in healthcare services provision can further contribute to inadequate medical treatments. (Line 52-53)
Reference
ILO, 2013. Domestic workers across the world: Global and regional statistics and the extent of legal protection. International Labor Office. Geneva.
Comment 5: Methods: why did the authors not perform a meta analysis on this dataset?
Table 2:
it would be very informative to add effect sizes to the table. Why not run a meta analysis?
Response: Thanks for your comments. Effect sizes were added where possible. The nature of this study is a scoping review, which is a descriptive study design. It is intended to determine the scope of a body of literature on a given topic. A scoping review does not undertake data synthesis (such as meta-analyses), which should only occur following assessment of risk of bias of included studies (Peters et al., 2022). Therefore, only descriptive analysis is performed in our study.
Reference
Peters, M., Godfrey, C., McInerney, P., Khalil, H., Larsen, P., Marnie, C., Pollock, D., Tricco, A. C., & Munn, Z. (2022). Best practice guidance and reporting items for the development of scoping review protocols. JBI evidence synthesis, 20(4), 953–968. https://doi.org/10.11124/JBIES-21-00242
Comment 6: Please correct the p-values: p<= 0.01. Are the values smaller (<) or equal (=) to 0.01? The p-values reported as p< 0.05 should also be entered correctly, please report the exact value, e.g., p = 0.025. Usually for such cases: p < 0.0001 the p-value should be reported this way (smaller than a specific value). The same for p-value > 0.05. It would be very informative to have the actual values.
Response: Thanks for your comments. We have checked the p-value for each included study. Some of them did not provide exact p-values, just reporting results like p < 0.01 or p < 0.05. All data shown in the table were extracted from the original studies.